# Photocatalytic Methane Conversion over Pd/ZnO Photocatalysts under Mild Conditions

Arthur Pignataro Machado, Saulo Amaral Carminati, Eliane Ribeiro Januário, Patricia Silvaino Ferreira, Jorge Moreira Vaz * and Estevam Vitorio Spinacé *

Instituto de Pesquisas Energéticas e Nucleares, IPEN/CNEN, Av. Prof. Lineu Prestes, 2242-Cidade Universitária, São Paulo 05508-000, SP, Brazil

* Correspondence: jmvaz@ipen.br (J.M.V.); espinace@ipen.br (E.V.S.); Tel.: +55-11-2810-5666 (E.V.S.)

**Abstract:** Here, Pd nanoparticles supported on ZnO were prepared by the alcohol-reduction and the borohydride-reduction methods, and their efficiency towards the photocatalytic conversion of methane under mild conditions were evaluated. The resulting Pd/ZnO photocatalysts were characterized by X-ray fluorescence, X-ray diffraction, X-ray photoelectron spectroscopy, UV–Vis, and transmission electron microscopy. The reactions were performed with the photocatalysts dispersed in water in a bubbling stream of methane under UV-light illumination. The products formed were identified and quantified by gas chromatography (GC-FID/TCD/MSD). The principal products formed were $C_2H_6$ and $CO_2$ with minor quantities of $C_2H_4$ and CO. No $H_2$ production was observed. The preparation methods influenced the size and dispersion of Pd nanoparticles on the ZnO, affecting the performance of the photocatalysts. The best performance was observed for the photocatalyst prepared by borohydride reduction with 0.5 wt% of Pd, reaching a $C_2H_6$ production rate of 686 $\mu mol \cdot h^{-1} \cdot g^{-1}$ and a $C_2H_6$ selectivity of 46%.

**Keywords:** methane conversion; zinc oxide; palladium nanoparticles; photocatalysis





## 1. Introduction

Methane ($CH_4$) is the main component of natural gas and has been recently used as a fuel due to its higher mass heat compared with other hydrocarbons; it is also an important raw material in many industrial chemical processes [1,2]. However, the $CH_4$ conversion in these processes require severe conditions of temperature and pressure in order to promote the breaking of the C-H bond, which tends to disrupt the $CH_4$ conversion processes, leading to carbon further undergoing oxidation towards undesired products.

The conversion of $CH_4$ into value-added multicarbon ($C_{2+}$) compounds under mild conditions has aroused worldwide interest over the past years, and emerges as an appealing approach to generate desired products while avoiding further oxidation of $CH_4$ and $C_2$ hydrocarbon products to $CO_2$. However, $CH_4$ conversion under mild conditions is a challenge given the energy required for its activation, which has received increasing attention, especially to produce ethane ($C_2H_6$) and ethylene ($C_2H_4$) [3–6].

The advantage of using photocatalytic reactions is the possibility of promoting even difficult reactions close to room temperature, since the photoenergy provides sufficient activation energy for the chemical reaction [7–10]. An interesting photocatalytic process was described by Li and collaborators [3,11], which has the advantage of combining the $CH_4$ conversion and the hydrogen ($H_2$) evolution from water, simultaneously. In this process, by employing $TiO_2$ as a photocatalyst, neither methane conversion nor $H_2$ production were observed. On the other hand, the deposition of Pt or Pd nanoparticles on $TiO_2$ greatly improved the production of ethane and hydrogen; this is because the $Pd/TiO_2$ photocatalyst was more selective for ethane production than the $Pt/TiO_2$ photocatalysts, while the latter was more active for $H_2$ production.

The conversion of methane by photocatalytic processes has been extensively investigated through different reaction conditions, and has mainly used semiconductor and hybrid metal/semiconductor materials as photocatalysts [12]. In this way, extensive work has been devoted to finding a prospective material that combines all the requirements for efficient mild and direct conversion of methane into high value-added products, which still remains an expressive challenge [5,13–19]. Arguably, ZnO has been one of the most commonly used wide-bandgap n-type semiconductor for photocatalytic $CH_4$ conversion [12]. It has been demonstrated that $Zn^+$-$O^-$ pairs act as surface active sites, where $O^-$ centers are responsible for breaking C-H bonds in $CH_4$, while $Zn^{2+}$ sites assist the C-C coupling. Nevertheless, pure ZnO is not efficient for $CH_4$ conversion, where the high recombination rate of photoinduced hole/electron pairs is the major drawback [12,20].

Herein, this work aimed to compare two methods of Pd deposition over ZnO nanoparticles, and to study their effects on the methane conversion in water in a flow reactor under mild conditions. In this way, it was possible to obtain Pd nanoparticles with different sizes and dispersions on the ZnO semiconductor and to observe the influence of these variables in the photocatalytic $CH_4$ conversion. This work may shed light on the design of modified ZnO photocatalysts to achieve higher efficiency towards the desired products.

## 2. Results and Discussion

### 2.1. Characterization of Catalysts

Table 1 presents the amount of Pd deposited on the ZnO surface determined by the wavelength dispersive X-ray fluorescence (WDXRF). Since the Pd/ZnO (1.00%) photocatalyst synthesized by BRM presented better photoactivity for $CH_4$ conversion than the material prepared by ARM, Pd/ZnO photocatalysts with different concentration were also produced through BRM in order to observe its influence on the photoactivity. For all samples, the Pd content is close to the nominal values.

**Table 1.** Pd content determined by wavelength dispersive X-ray fluorescence.

| Photocatalyst | Photocatalyst Composition | Method of Synthesis | Pd Content (wt. %) | |
|---|---|---|---|---|
| | | | Nominal | WDXRF |
| Sample A | Pd (1.00%)/ZnO | ARM | 1.00 | 1.22 |
| Sample B | Pd (1.00%)/ZnO | BRM | 1.00 | 1.13 |
| Sample C | Pd (0.50%)/ZnO | BRM | 0.50 | 0.55 |
| Sample D | Pd (0.25%)/ZnO | BRM | 0.25 | 0.44 |

Figure 1a shows the XRD patterns of commercial ZnO and the samples A and B. According to the crystallography open database (COD), the (100), (002), (101), (102), (110), (103), (200), (112), (201), and (204) diffraction peaks belong to the hexagonal crystal structure of ZnO (CARD 00-900-4180). Given the low Pd content of the samples, no peaks different from those found in the ZnO matrix were observed in the XRD patterns of the as-prepared photocatalysts. This can be assigned to the broadening of Pd peaks caused by the small size of the nanoparticles, as well as their low concentration in the material [3,11].

The synthesized photocatalysts were also characterized by UV–vis diffuse reflectance spectroscopy (DRS), and the samples spectra are shown in Figure 1b. It is possible to notice that the ZnO photocatalyst activation occurs around 380 nm of UV light. For the bandgap energy calculations, the reflectance spectra were converted using the Kulbelk–Munk function as the ideal model for relating the reflectance and absorbance in powder samples [21]. The Tauc plot method was performed considering the direct allowed transition of ZnO [22]. The pristine semiconductor presented a bandgap energy of 3.31 eV, and no change was observed in the bandgap of Pd-containing samples compared to pure semiconductor. This was related to the fact that Pd is only deposited on the surface of ZnO, and no structural change occurs in the bulk material, which is in agreement with the XRD results obtained.

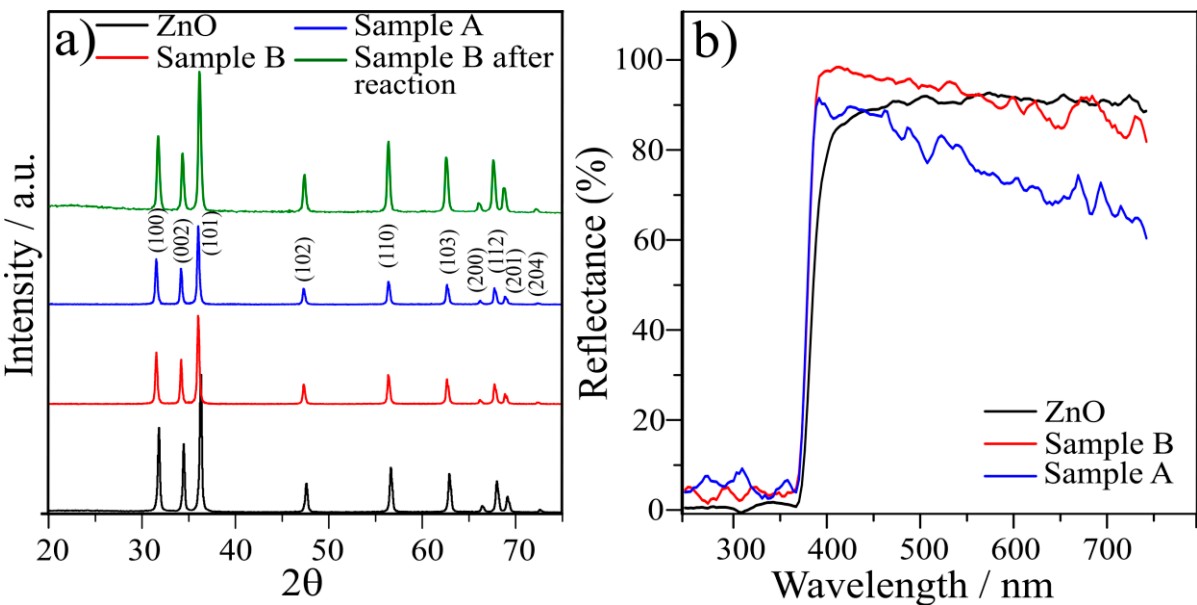

**Figure 1.** (**a**) XRD patterns and (**b**) diffuse reflectance UV–vis spectra of the photocatalysts.

Figure 2 shows the TEM images of pure ZnO. Nanorods and hexagonal nanoparticles varying their size in the range of 20 to 200 nm could be identified, but no uniform size and morphology were present among the ZnO nanoparticles' structure.

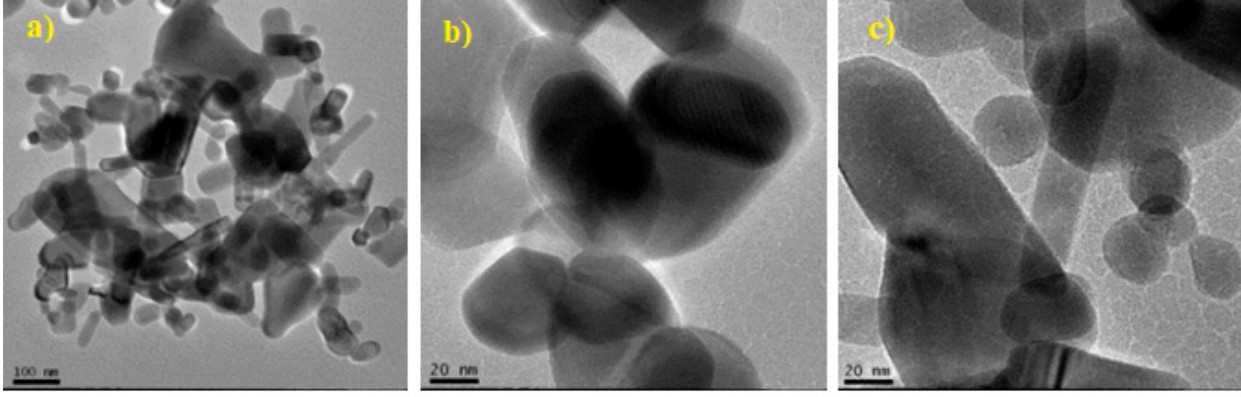

**Figure 2.** (**a**–**c**) TEM images of pure ZnO nanoparticles.

Figure 3 shows the TEM images of samples A and B synthesized by ARM (Figure 3a,b) and BRM (Figure 3c,d), respectively. Comparing the images of pure ZnO with those of the synthesized photocatalysts, the appearance of black dots on the surface of ZnO, together with the data obtained by WDXRF, confirm that both methods of synthesis were successful for the deposition of Pd nanoparticles on the ZnO semiconductor surface.

Importantly, by analyzing the TEM images of Sample A (Figure 3a,b), it can be observed that the Pd nanoparticles are more agglomerated when compared to Sample B (Figure 3c,d). The average nanoparticles sizes are greater in Sample A (Figure 4a) than Sample B (Figure 4b). Thus, the ARM method produced larger and more agglomerated Pd nanoparticles, while the BRM method led to smaller and more dispersed Pd nanoparticles on the ZnO surface. Such differences between the materials produced by the different methods could be associated with the presence of the citric acid as a dispersing agent in the BRM [23].

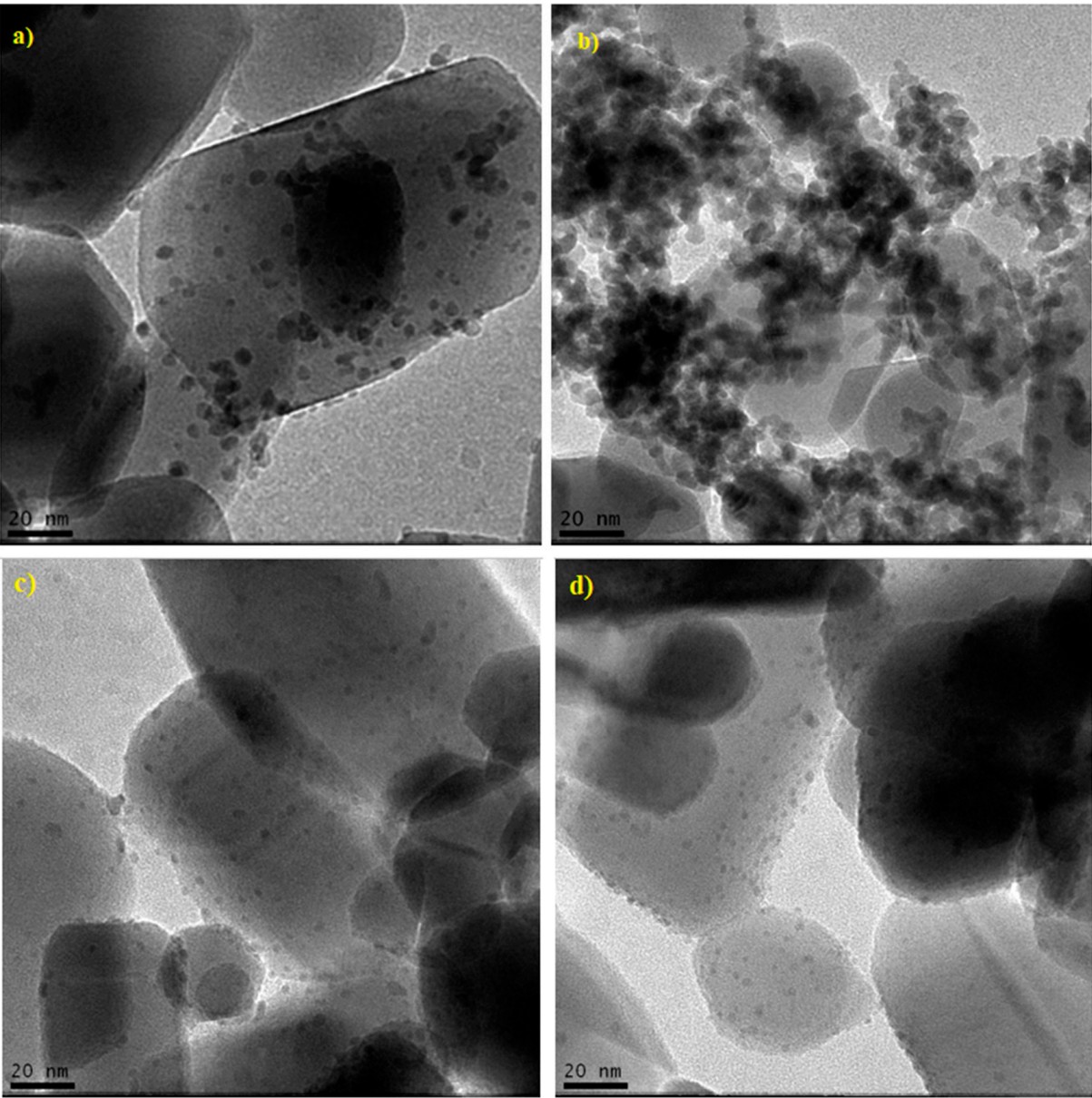

**Figure 3.** Transmission electron microscopy images of (**a**,**b**) Sample A and (**c**,**d**) Sample B.

The XPS analysis was used to determine the chemical state of the elements in the samples before and after irradiation, as shown in Figure 5. Figure 5a shows the Zn $2p_{3/2}$ and $2p_{1/2}$ region of Sample B before irradiation, corresponding to 1021.41 eV and 1044.53 eV binding energies, respectively. For the same sample after photocatalytic testing, the binding energies show a slight shift to 1021.50 eV and 1044.62 eV, respectively [24]. The difference of ~23.1 eV between them indicates the presence of Zn in a +2 oxidation state even before or after the irradiation experiments. The peaks located at 335.48 and 340.76 eV are assigned to Pd 3d in its $Pd^0$ chemical state, with no +2 and +4 oxidation states, respectively. After irradiation, these peaks shifted to 335.14 and 340.38 eV, respectively, and the presence of the Pd +2 species was observed, although most of the Pd existed in oxidation state 0. The XRD (Figure 1a) also confirms that the ZnO structure was preserved after irradiation, showing that the photocatalyst has good stability under reaction conditions.

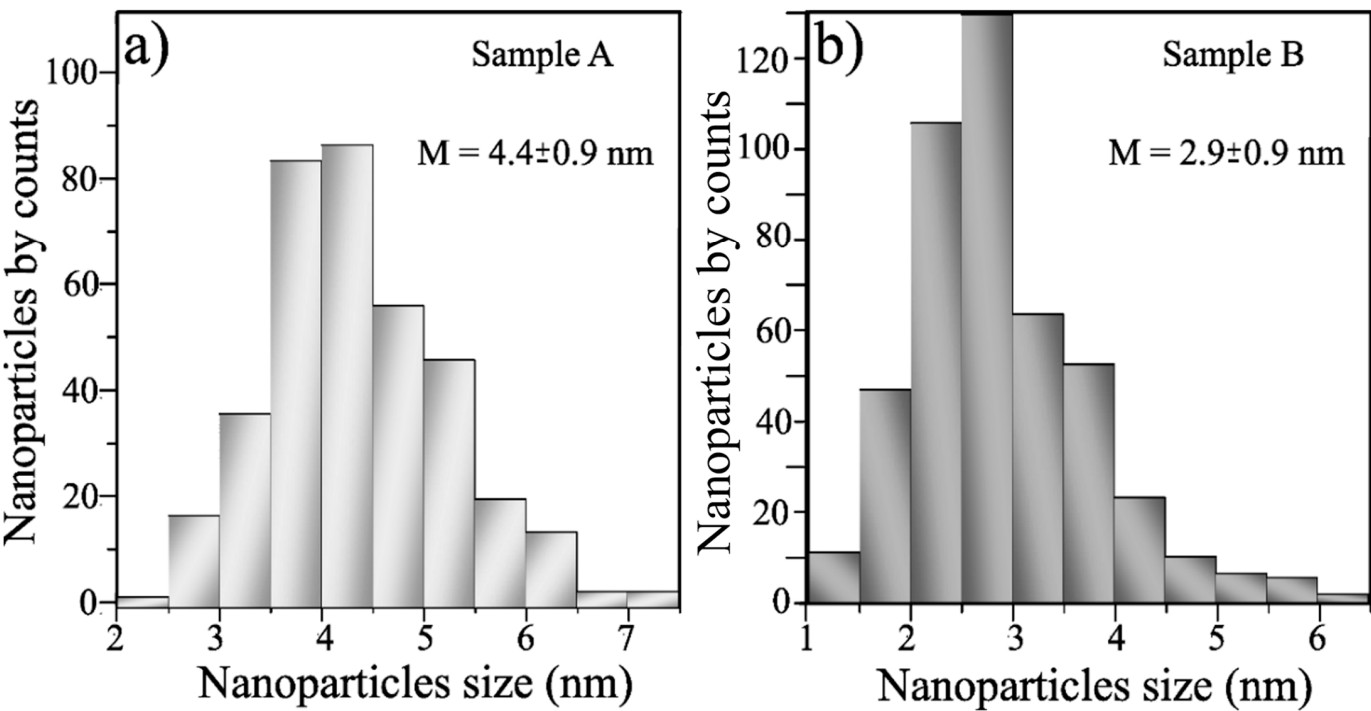

**Figure 4.** Histogram plots of nanoparticle size (nm) and count of (**a**) Sample A and (**b**) Sample B from the TEM images.

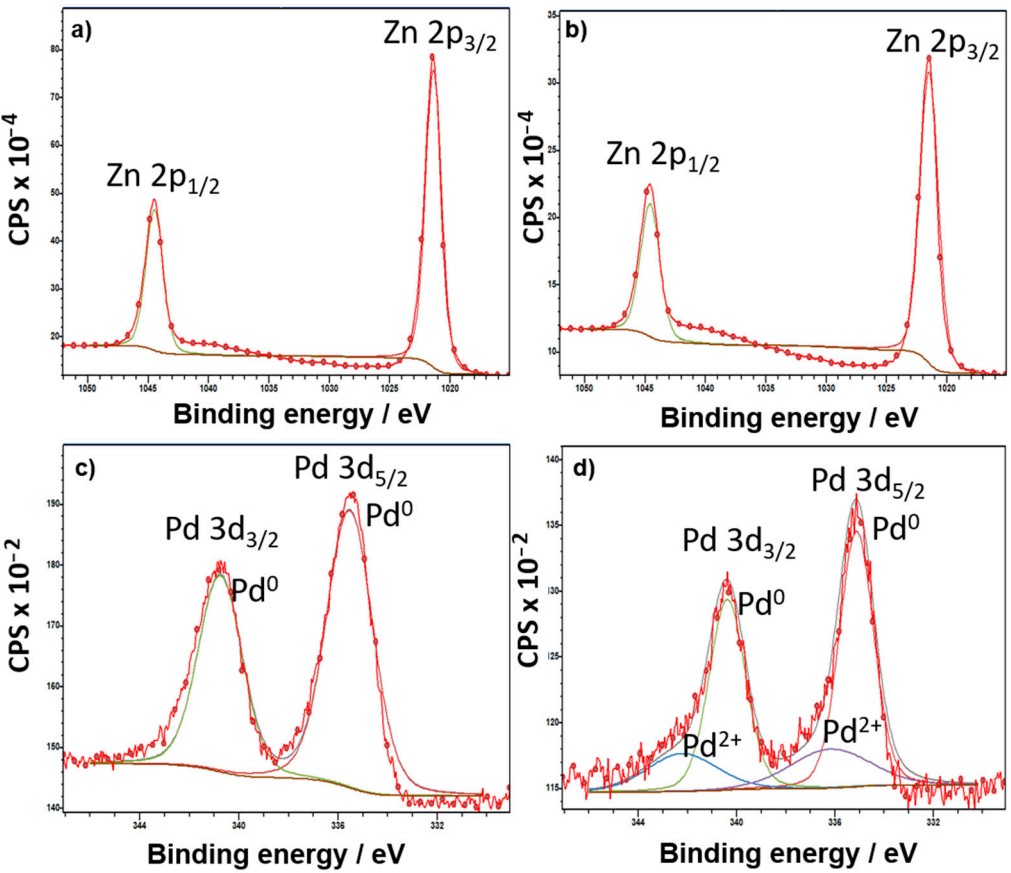

**Figure 5.** Zn 2p spectrum of Sample B (**a**) before and (**b**) after irradiation; Pd 3d spectrum of Sample B (**c**) before and (**d**) after irradiation.

## 2.2. Photocatalytic Tests

The photocatalytic activity of ZnO and Pd/ZnO photocatalysts are shown in Figure 6, where the graphs represent the percentages of the different products present in the $CH_4$ flow rate on the ordinate axis and the chromatographic injections performed during the experiments on the abscissa axis. Among the 12 chromatographic injections performed during the photoactivity analysis, the first was discarded. The subsequent two injections were collected while the light source remained switched off, whereas injections three to nine were performed upon illumination. Before the last two injections, the light was turned off again. Initially, a blank experiment (Figure 6a), using only ultrapure water and $CH_4$, was performed in order to observe possible photochemical reactions that may occur in the system. In this case, a very slight increase in the production of $CO_2$, $C_2H_6$, and $C_2H_4$ was observed upon photoirradiation. This was ascribed to the photochemical reactions that $CH_4$ can undergo, although the amounts of evolved products were extremely low. The addition of pure ZnO to the system (Figure 6b) increased the $CO_2$ production by almost 12-fold upon photoirradiation, while small amounts of $C_2H_6$ and CO were formed. This behavior was associated with the photocatalytic activity of the ZnO when photoexcited across the band gap, promoting the formation of electron ($e^-$)/hole ($h^+$) pairs. These species can then promote the formation of methyl radicals ($\bullet CH_3$) through the direct interaction of $CH_4$ with holes, or indirectly through its interaction with other radicals resulting from photocatalytic reactions, such as hydroxyl radicals ($\bullet OH$) produced from water. The formation of $\bullet CH_3$ is a crucial step in the photocatalytic conversion of $CH_4$ to $C_2H_6$, as it allows coupling reactions between these radicals to occur [19]. The Pd/ZnO photocatalyst synthesized by ARM (Sample A), when compared to pure ZnO, showed enhanced photoactivity, leading to an increase in the $CO_2$ and $C_2H_6$ production, as it can be seen in Figure 6c. The Pd/ZnO (1.0%) photocatalyst synthesized by BRM (Sample B) showed enhanced photoactivity compared to Sample A and its products' formation can be seen in Figure 6d.

The products' formation rates ($\mu mol \cdot h^{-1} \cdot g^{-1}$) of the blank experiment using ZnO and Pd/ZnO photocatalysts are shown in Table 2.

**Table 2.** Products' formation rates of ZnO and Pd (1.0%)/ZnO photocatalysts synthesized by ARM and BRM methods.

| Photocatalyst | Products' Formation Rate ($\mu mol \cdot h^{-1} \cdot g^{-1}$) | | | | Selectivity (%) | | | |
|---|---|---|---|---|---|---|---|---|
| | $CO_2$ | $C_2H_4$ | $C_2H_6$ | CO | $CO_2$ | $C_2H_4$ | $C_2H_6$ | CO |
| Blank | 9 | - | - | - | 100 | - | - | - |
| ZnO Sigma-Aldrich | 113 | - | 5 | 7 | 90.4 | - | 4.0 | 5.6 |
| Sample A | 214 | 2 | 56 | 4 | 77.5 | 0.7 | 20.3 | 1.5 |
| Sample B | 336 | 14 | 291 | 3 | 52.2 | 2.2 | 45.2 | 0.4 |

The addition of Pd to the ZnO semiconductor increased the products; formation rates and strongly modified the selectivity when compared to the bare ZnO photocatalyst. In addition, it is important to highlight that the size and distribution of Pd nanoparticles on ZnO also influence the quantity and selectivity of the evolved products. The best performance was observed for Sample B with a $C_2H_6$ formation rate of four times greater than that of Sample A and with a $C_2H_6$ selectivity of 45% while, for Sample A, it was only 20%.

Therefore, Pd/ZnO photocatalysts with different Pd wt% loadings were prepared by BRM. The products' formation rate ($\mu mol \cdot h^{-1} \cdot g^{-1}$) are shown in Table 3.

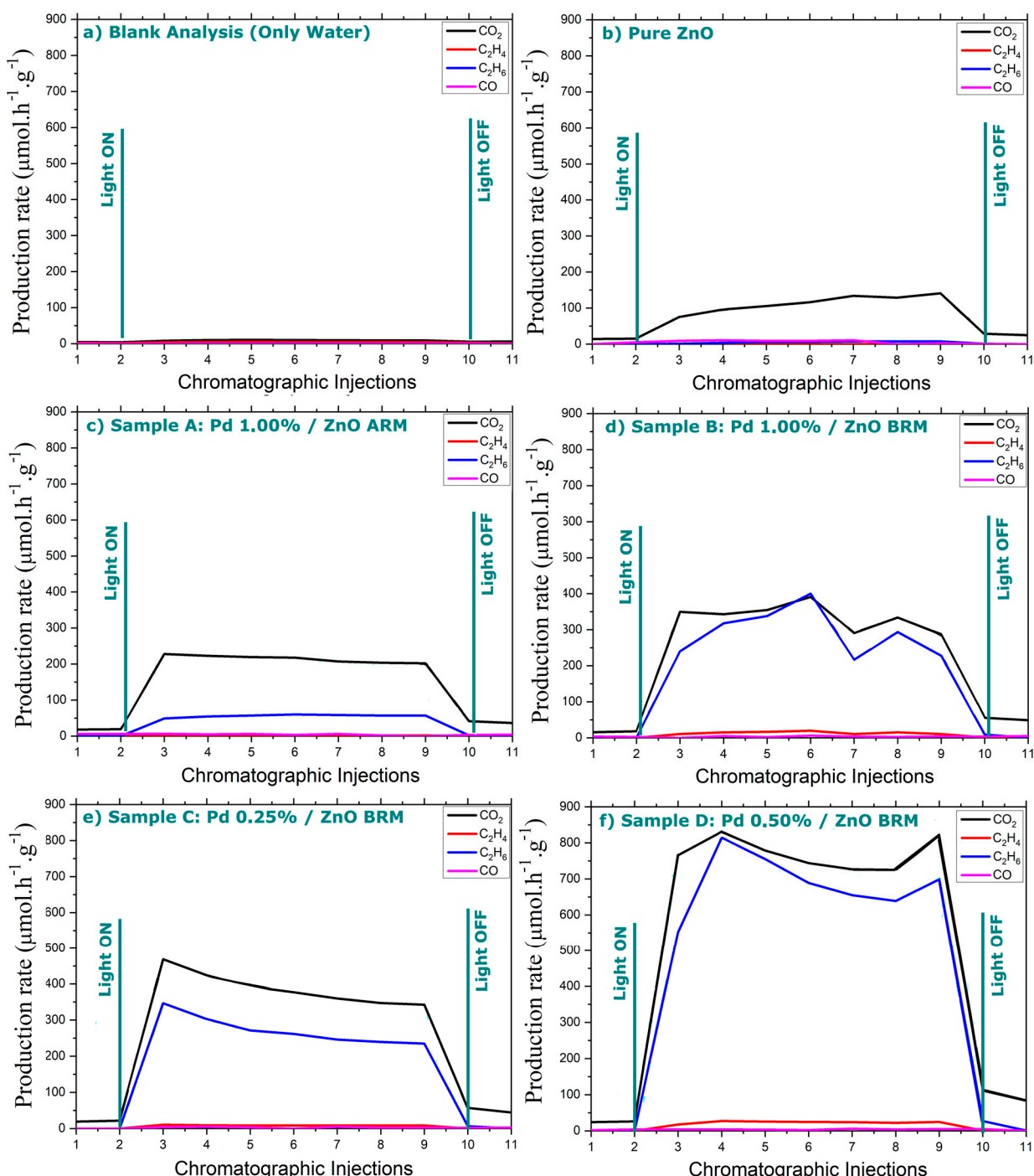

**Figure 6.** Photocatalytic products' formation profiles for (**a**) blank (**b**) pure commercial ZnO, (**c**) Sample A, (**d**) Sample B, (**e**) Sample C, and (**f**) Sample D.

From the data in Table 3, it is noticeable that the amount of Pd deposited on the ZnO surface affects the products' formation rates, with the amount of 0.5 wt% (Sample C) showing the best performance. It is worth mentioning that an additional increase in the Pd charge causes a decrease in the performance and probably contributes to the recombination process of the charge carriers instead of promoting charge separation [25]. In this manner, Sample C showed a $C_2H_6$ formation rate of 686 $\mu mol \cdot h^{-1} \cdot g^{-1}$, a $C_2H_4$ formation rate of 24 $\mu mol \cdot h^{-1} \cdot g^{-1}$, and a $C_2H_6$ selectivity of 46%. For all Pd/ZnO samples prepared through BRM, a $C_2H_6$: $CO_2$ molar ratio of approximately 1:1 was observed. Using Pd/$TiO_2$ as the photocatalyst, Li and Yu [3] achieved a $C_2H_6$ formation rate of

55 $\mu mol \cdot h^{-1} \cdot g^{-1}$ with a $C_2H_6$: $CO_2$ molar ratio of approximately 2.5:1 and a $H_2$ production rate of 122 $\mu mol \cdot h^{-1} \cdot g^{-1}$. Curiously, no $H_2$ production was observed for ZnO and for all Pd/ZnO photocatalysts. On the other hand, when Pd nanoparticles were supported on $TiO_2$ and $Ga_2O_3$ by ARM or BRM methods and tested for $CH_4$ conversion in the same conditions, the formation of $H_2$ was observed in appreciable quantities (in the range of 10 to 30 $mmol \cdot h^{-1} \cdot g^{-1}$). A similar result was recently described for the photocatalytic non-oxidative coupling of methane using ZnO as the photocatalyst, which showed $C_2H_6$ production but no production of $H_2$, while $H_2$ formation was observed for $TiO_2$ and $Ga_2O_3$ photocatalysts. The authors suggest that ZnO was possibly reduced by $H_2$ upon photoirradiation [26].

**Table 3.** Products' formation rates of Pd/ZnO photocatalysts synthesized by BRM with different Pd wt% loading.

| Photocatalyst | Products' Formation Rates ($\mu mol \cdot h^{-1} \cdot g^{-1}$) | | | | Selectivity (%) | | | |
|---|---|---|---|---|---|---|---|---|
| | $CO_2$ | $C_2H_4$ | $C_2H_6$ | CO | $CO_2$ | $C_2H_4$ | $C_2H_6$ | CO |
| Sample B | 336 | 14 | 291 | 3 | 52.2 | 2.2 | 45.2 | 0.4 |
| Sample C | 770 | 24 | 686 | 4 | 51.9 | 1.6 | 46.2 | 0.3 |
| Sample D | 387 | 9 | 272 | 3 | 57.7 | 1.3 | 40.5 | 0.5 |

Li and Yu [3] proposed a mechanism for Pd/$TiO_2$ photocatalysts. Initially, the water was activated by holes, forming the •OH radicals, which reacted with $CH_4$ molecules, forming •$CH_3$ radicals that were responsible for the formation of the $C_2$ products. It was also inferred that the $H_2$ production comes primarily from water molecules and that Pd acts as an electron trap to avoid its recombination with the hole, and as a center for $CH_4$ activation. Recently, the generation of •OH radicals in metal nanoparticles (Pt, Pd, Au, or Ag) supported on $TiO_2$ and ZnO was measured by photoluminescence [6]. The authors showed that metal/$TiO_2$ photocatalysts were more efficient than metal/ZnO in producing •OH radicals [6]. Based on these results, it is possible that in our system using Pd/ZnO photocatalysts, the $CH_4$ is preferably activated directly by the holes rather than indirectly by •OH radicals, as shown in Figure 7.

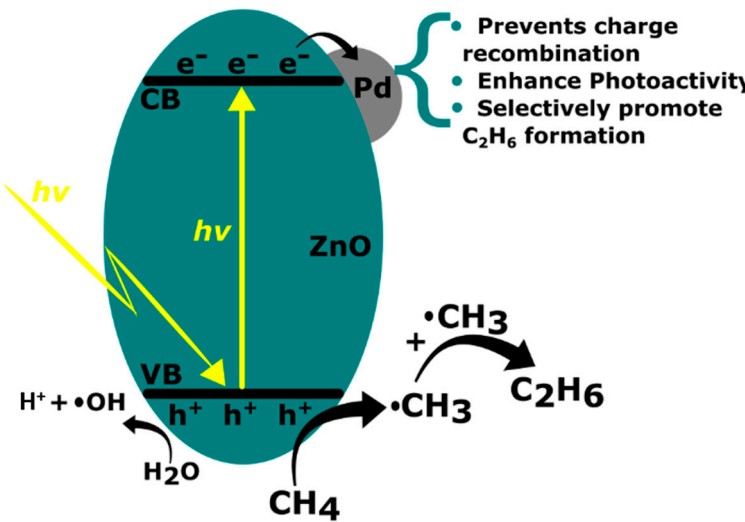

**Figure 7.** Proposed mechanism for $C_2H_6$ production with a Pd/ZnO photocatalyst.

Our results showed that the Pd/ZnO photocatalyst with smaller Pd nanoparticles sizes and good dispersion on a ZnO semiconductor seems to contribute to a more efficient separation of photogenerated charges (holes and electrons), contributing to a greater efficiency in the system, which enhances the $C_2H_6$ formation. Recently, a high-performance

Pd/TiO$_2$ photocatalyst, where TiO$_2$ was decorated with Pd single atoms highly dispersed on TiO$_2$, was described for the photocatalytic non-oxidative conversion of methane to C$_2$H$_6$, resulting in a production rate of 910 μmol·h$^{-1}$·g$^{-1}$ and suppressing the over-oxidation to CO$_2$ [27]. Compared to the photocatalyst in which Pd nanoparticles were dispersed on TiO$_2$, the single-atom photocatalyst was much more active, demonstrating that the size and the dispersion of the metallic atoms on the semiconductor can strongly influence the activity and selectivity of these materials [27].

## 3. Materials and Methods

### 3.1. Photocatalysts Preparation

All chemicals were of analytical grade and used without any further purification. The Pd/ZnO photocatalysts were prepared by the alcohol-reduction method (ARM) [28,29] and by the borohydride reduction method (BRM) [30].

For both methods of synthesis, the prepared Pd nanoparticles were deposited over nanosized commercial ZnO obtained from Saint Louis—USA Sigma-Aldrich ($\leq$100 nm particle size). An aqueous solution of sodium tetrachloropalladate (Na$_2$PdCl$_4$·3H$_2$O) was used as a Pd precursor.

### 3.1.1. Alcohol-Reduction Method (ARM)

The ARM utilizes ethylene glycol (EG) as a reducing agent [28,29]. Briefly, proper amounts of Pd precursor and ZnO were dispersed in an aqueous solution containing EG in a ratio of 3:1 EG/H$_2$O. The reaction was refluxed (175 °C) under vigorous stirring for 1 h. The solid was separated by centrifugation, washed several times with distilled water, and dried at 80 °C. The resulting material was ground to a powdery appearance.

### 3.1.2. Borohydride Reduction Method (BRM)

The BRM utilizes sodium borohydride as a reducing agent and sodium citrate as a dispersing agent [30]. The ZnO was dispersed in an aqueous solution containing a proper amount of Pd precursor and sodium citrate (Pd:citrate ratio 1:3). An aqueous solution of sodium borohydride was dropped into the mixture under vigorous stirring at room temperature. The reaction was maintained under stirring for 24 h. The dispersed solid was separated by centrifugation, washed several times with distilled water, and dried at 80 °C. The resulting material was ground to a powdery appearance.

### 3.2. Characterizations

The Pd content (wt%) was determined by wavelength dispersive X-ray fluorescence (WDXRF), performed using a Rigaku Supermini200 spectrometer with a 50 kV Palladium anode X-ray tube with 200 W of potency and a zirconium bean filter. Then, UV–Vis diffuse reflectance spectroscopy analysis was carried out using a Varian Cary50 UV-Vis Spectrophotometer with a xenonium lamp and barium sulfate (BaSO$_4$) as a blank pattern. The X-ray diffraction analysis was performed using a Bruker D8 Advance 3 kW instrument using a copper tube and a scintillation detector. Transmission electron microscopy images of the synthesized materials were obtained from a 200 keV JEOL JEM 2010. The X-ray photoelectron spectroscopy (XPS) experiments were carried out with K-alpha surface analysis (Thermo Scientific, Waltham, MA, USA) equipment with an Al-K$\alpha$ X-ray source (1486.6 eV) and a flood gun.

### 3.3. Photocatalytic Tests

The photocatalytic activity measurements were carried out in a 250 mL Ace photochemical reactor coupled to the GC-FID/TCD/MSD system. The photocatalysts were dispersed in 250 mL of ultrapure water and CH$_4$ was bubbled through in a flow ratio of 25 mL min$^{-1}$, while a 450 W Hg lamp was used as a light source. In addition, two cooling systems were used, one coupled to a condenser at the output of the photoreactor that was connected to the GC system to condense the water (15 °C), and the other for cooling the Hg

lamp (40 °C). In this way, the photocatalytic reactions were carried out at a temperature close to 60 °C.

The gas chromatographic (GC) model utilized was Agilent 7890B coupled to MSD 5977B. The equipment has a thermal conductivity detector (TCD), methanizer (MET), and flame ionization detector (FID), as well as a quadrupole mass spectrometer detector (MSD). Two different columns were used in order to separate the reaction products, namely a plot U and a molecular sieve 5 Å column. Twelve injections were performed in a total of 7 h of analysis, each one of 33 min. The first 3 injections took place with the light switched off, while injections 4 to 10 took place with the light switched on; in the last 2 injections the light was turned off again, so it was possible to monitor the influence of light on the system. Prior to testing the activity of the catalyst, calibration curves were produced to quantify $CO_2$, $C_2H_4$, $C_2H_6$, $C_3H_8$, $C_4H_{10}$, $H_2$, $CH_4$, and CO. The detection limits were 0.001% for $CO_2$ and $C_2$-$C_4$; 0.008% for $CH_4$ and CO, and 0.3% for $H_2$.

Two certified gas mixtures containing some of the expected products (carbon dioxide, ethane, ethene, propane, butane, carbon monoxide) at different known concentrations were used to build a calibration curve in order to analyze the products formed during the photocatalytic reaction.

The selectivity was calculated according to the following equation:

$$Product\ selectivity = \frac{n\ product}{n\ total\ of\ the\ products\ formed} \times 100\% \tag{1}$$

where $n$ represents the molar amounts.

## 4. Conclusions

Here, $Pd/ZnO$ photocatalysts dispersed in water in a bubbling stream of methane under UV-light illumination were shown to be active for $CH_4$ conversion. The main products formed were $C_2H_6$ and $CO_2$, with minor quantities of $C_2H_4$ and CO; however, no $H_2$ production was observed. The photocatalyst preparation methods influenced the size and dispersion of Pd nanoparticles on the ZnO support, playing a pivotal role in the quantity and selectivity of the products formed. The $Pd/TiO_2$ photocatalysts with smaller Pd particle sizes, good dispersion, and an optimal Pd content were shown to be more active and selective for $C_2H_6$ production.

**Author Contributions:** A.P.M. performed the catalyst synthesis, characterization, and photocatalytic tests; E.R.J. and P.S.F. participated in the catalyst's preparation, characterization, and photocatalytic tests; A.P.M. and S.A.C. analyzed the data and wrote the paper; J.M.V. and E.V.S. designed the study and reviewed the paper. All authors have read and agreed to the published version of the manuscript.

**Funding:** The authors gratefully acknowledge support from FAPESP (São Paulo Research Foundation Grant numbers 2017/11937-4, 2018/04596-9 and 2018/04595-2), Shell, and the strategic importance of the support given by ANP (Brazil's National Oil, Natural Gas and Biofuels Agency) through R&D levy regulation. Furthermore, S.A.C. thanks FAPESP for the fellowship (Grant number 2021/01896-4), and E.V.S. thanks the Brazilian National Council for Scientific Development (CNPq, Grant number 305622/2020-0).

**Institutional Review Board Statement:** Not applicable.

**Informed Consent Statement:** Not applicable.

**Data Availability Statement:** All relevant data are included in the paper.

**Acknowledgments:** We thank LNNano (Laboratório de Microscopia Eletrônica) for the XPS measurements.

**Conflicts of Interest:** The authors declare no conflict of interest.

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
