# Peer review of "Photocatalytic Methane Conversion over Pd/ZnO Photocatalysts under Mild Conditions"

_methane, doi:10.3390/methane2010003_

Round 1
Reviewer 1 Report
This is an interesting paper on using Pd-ZnO for light-driven methane conversion. I suggest the paper for publication after proper revisions.
1. The selectivity for all products must be added. Carbon mass balance should be given.
2. Please provide the evidence for detecting the methane radicals.
3. Recent progress in this field is missing.
Catalysis Communications, 2021, 159, 106346
Author Response
The file was attached

Reviewer 2 Report
The manuscript entitled “Photocatalytic Methane Conversion over Pd/ZnO Photocatalysts Under Mild Conditions” submitted by Machado et al. in Methan needs to fix the following points:
1. There are some mistakes or typo errors in the text. There is some unnecessary space provided by them in text. The authors should go carefully through the manuscript and fix.
2. HRTEM data should be provided instead of additional TEM images.
3. XPS fitting is missing.
4. I have main concern regarding the stability of the catalysts in term of the structural and morphological changes. Stability of the catalysts should be checked after catalytic reactions. XRD or XPS and microscopic studies of the recovered catalysts are highly recommended.
Author Response
the file was attached

Round 2
Reviewer 1 Report
I recommend the revsied paper to publish now.
Reviewer 2 Report
I do not understand that why so many TEM images at different scale bar are given. It is better to provide High Resolution TEM image instead of TEM images at various magnifications.
Stability of the catalysts should be checked after catalytic reactions. XRD or XPS and microscopic studies of the recovered catalysts are highly recommended. This is the main concern regarding the stability of the catalysts in term of the structural and morphological changes.
